# Dual-Agent GANs for Photorealistic and Identity Preserving Profile Face Synthesis

Jian Zhao[1,2]*†    Lin Xiong[3]    Karlekar Jayashree[3]    Jianshu Li[1]    Fang Zhao[1]
Zhecan Wang[4]†    Sugiri Pranata[3]    Shengmei Shen[3]
Shuicheng Yan[1,5]    Jiashi Feng[1]
[1]National University of Singapore    [2]National University of Defense Technology
[3] Panasonic R&D Center Singapore    [4] Franklin. W. Olin College of Engineering
[5] Qihoo 360 AI Institute
{zhaojian90, jianshu}@u.nus.edu    {lin.xiong, karlekar.jayashree, sugiri.pranata, shengmei.shen}@sg.panasonic.com
zhecan.wang@students.olin.edu    {elezhf, eleyans, elefjia}@u.nus.edu

## Abstract

Synthesizing realistic profile faces is promising for more efficiently training deep pose-invariant models for large-scale unconstrained face recognition, by populating samples with extreme poses and avoiding tedious annotations. However, learning from synthetic faces may not achieve the desired performance due to the discrepancy between distributions of the synthetic and real face images. To narrow this gap, we propose a **D**ual-**A**gent **G**enerative **A**dversarial **N**etwork (DA-GAN) model, which can improve the realism of a face simulator's output using *unlabeled* real faces, while preserving the identity information during the realism refinement. The dual agents are specifically designed for distinguishing real *v.s.* fake and identities simultaneously. In particular, we employ an off-the-shelf 3D face model as a simulator to generate profile face images with varying poses. DA-GAN leverages a fully convolutional network as the generator to generate high-resolution images and an auto-encoder as the discriminator with the dual agents. Besides the novel architecture, we make several key modifications to the standard GAN to preserve pose and texture, preserve identity and stabilize training process: (i) a pose perception loss; (ii) an identity perception loss; (iii) an adversarial loss with a boundary equilibrium regularization term. Experimental results show that DA-GAN not only presents compelling perceptual results but also significantly outperforms state-of-the-arts on the large-scale and challenging NIST IJB-A unconstrained face recognition benchmark. In addition, the proposed DA-GAN is also promising as a new approach for solving generic transfer learning problems more effectively. DA-GAN is the foundation of our submissions to NIST IJB-A 2017 face recognition competitions, where we won the $1^{st}$ places on the tracks of verification and identification.

## 1 Introduction

Unconstrained face recognition is a very important yet extremely challenging problem. In recent years, deep learning techniques have significantly advanced large-scale unconstrained face recognition (8; 19; 27; 34; 29; 16), arguably driven by rapidly increasing resource of face images. However, labeling huge amount of data for feeding supervised deep learning algorithms is undoubtedly expensive and time-consuming. Moreover, as often observed in real-world scenarios, the pose distribution of available face recognition datasets (*e.g.*, IJB-A (15)) is usually unbalanced and has long-tail with

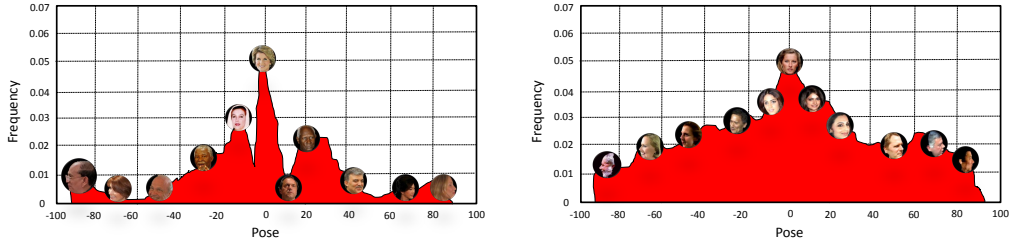

(a) Extremely unbalanced pose distribution.  (b) Well balanced pose distribution with DA-GAN.

Figure 1: Comparison of pose distribution in the IJB-A (15) dataset w/o and w/ DA-GAN.

large pose variations, as shown in Figure. 1a. This has become a main obstacle for further pushing unconstrained face recognition performance. To address this critical issue, several research attempts (32; 31; 35) have been made to employ synthetic profile face images as augmented extra data to balance the pose variations.

However, naively learning from synthetic images can be problematic due to the distribution discrepancy between synthetic and real face images—synthetic data is often not realistic enough with artifacts and severe texture losses. The low-quality synthesis face images would mislead the learned face recognition model to overfit to fake information only presented in synthetic images and fail to generalize well on real faces. Brute-forcedly increasing the realism of the simulator is often expensive in terms of time cost and manpower, if possible.

In this work, we propose a novel **D**ual-**A**gent **G**enerative **A**dversarial **N**etwork (DA-GAN) for profile view synthesis, where the dual agents focus on discriminating the realism of synthetic profile face images from a simulator using unlabled real data and perceiving the identity information, respectively. In other words, the generator needs to play against a real–fake discriminator as well as an identity discriminator simultaneously to generate high-quality faces that are really useful for unconstrained face recognition.

In our method, a synthetic profile face image with a pre-specified pose is generated by a 3D morphable face simulator. DA-GAN takes this synthetic face image as input and refines it through a conditioned generative model. We leverage a **F**ully **C**onvolutional **N**etwork (FCN) (17) that operates on the pixel level as the generator to generate high-resolution face images and an auto-encoder network as the discriminator. Different from vanilla GANs, DA-GAN introduces an auxiliary discriminative agent to enforce the generator to preserve identity information of the generated faces, which is critical for face recognition application. In addition, DA-GAN also imposes a pose perception loss to preserve pose and texture. The refined synthetic profile face images present photorealistic quality with well preserved identity information, which are used as augmented data together with real face images for pose-invariant feature learning. For stabilizing the training process of such dual-agent GAN model, we impose a boundary equilibrium regularization term.

Experimental results show that DA-GAN not only presents compelling perceptual results but also significantly outperforms state-of-the-arts on the large-scale and challenging **N**ational **I**nstitute of **S**tandards and **T**echnology (NIST) **I**ARPA **J**anus **B**enchmark **A** (IJB-A) (15) unconstrained face recognition benchmark. DA-GAN leads us to further win the $1^{st}$ places on verification and identification tracks in the NIST IJB-A 2017 face recognition competitions. This strong evidence shows that our "recognition via generation" framework is effective and generic, and we expect that it benefits for more face recognition and transfer learning applications in the real world.

Our contributions are summarized as follows.

- We propose a novel **D**ual-**A**gent **G**enerative **A**dversarial **N**etwork (DA-GAN) for photorealistic and identity preserving profile face synthesis even under extreme poses.
- The proposed dual-agent architecture effectively combines prior knowledge from data distribution (adversarial training) and domain knowledge of faces (pose and identity perception losses) to exactly recover the lost information inherent in projecting a 3D face into the 2D image space.
- We present qualitative and quantitative experiments showing the possibility of a "recognition via generation" framework and achieve the top performance on the challenging NIST IJB-A (15) unconstrained face recognition benchmark without extra human annotation efforts

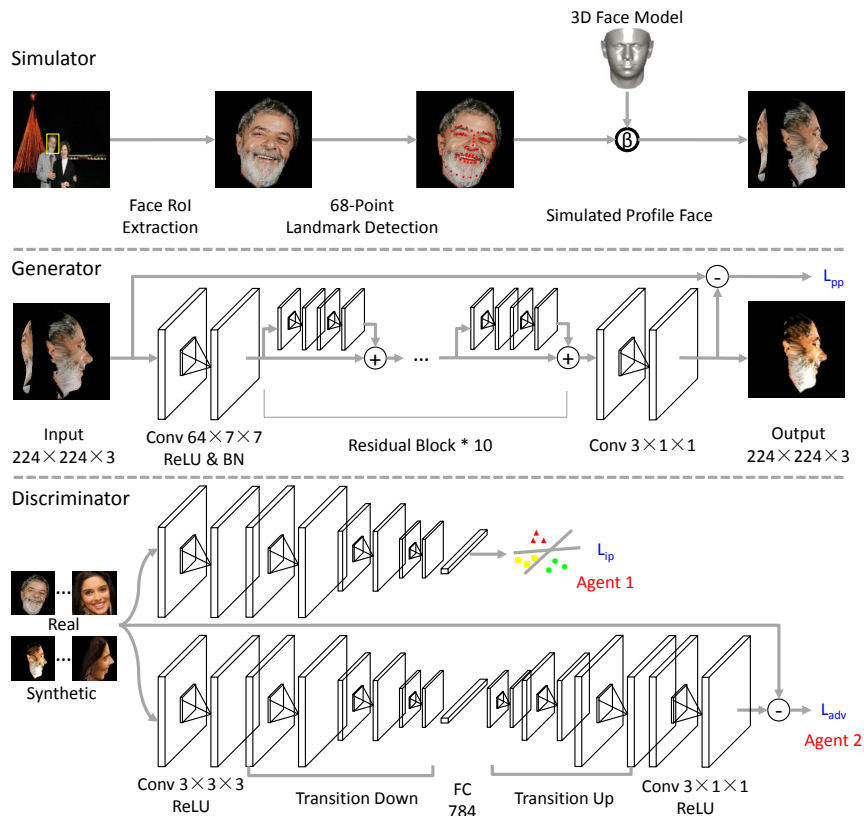

Figure 2: Overview of the proposed DA-GAN architecture. The simulator (upper panel) extracts face RoI, localizes landmark points and produces synthesis faces with arbitrary poses, which are fed to DA-GAN for realism refinement. DA-GAN uses a fully convolutional skip-net as the generator (middle panel) and an auto-encoder as the discriminator (bottom panel). The dual agents focus on both discriminating real *v.s.* fake (minimizing the loss $\mathcal{L}_{\mathrm{adv}}$) and preserving identity information (minimizing the loss $\mathcal{L}_{\mathrm{ip}}$). Best viewed in color.

by training deep neural networks on the refined face images together with real images. To our best knowledge, our proposed DA-GAN is the first model that is effective for automatically generating augmented data for face recognition in challenging conditions and indeed improves performance. DA-GAN won the $1^{st}$ places on verification and identification tracks in the NIST IJB-A 2017 face recognition competitions.

## 2 Related works

As one of the most significant advancements on the research of deep generative models (14; 26), GAN has drawn substantial attention from the deep learning and computer vision community since it was first introduced by Goodfellow *et al.* (10). The GAN framework learns a generator network and a discriminator network with competing loss. This min-max two-player game provides a simple yet powerful way to estimate target distribution and to generate novel image samples. Mirza and Osindero (21) introduce the conditional version of GAN, to condition on both the generator and discriminator for effective image tagging. Berthelot *et al.* (2) propose a new **B**oundary **E**quilibrium **GAN** (BE-GAN) framework paired with a loss derived from the Wasserstein distance for training GAN, which derives a way of controlling the trade-off between image diversity and visual quality. These successful applications of GAN motivate us to develop profile view synthesis methods based on GAN. However, the generator of previous methods usually focus on generating images based on a random noise vector or conditioned data and the discriminator only has a single agent to distinguish real *v.s.* fake. Thus, in contrast to our method, the generated images do not have any discriminative information that can be used for training a deep learning based recognition model. This separates us well with previous GAN-based attempts.

Moreover, differnet from previous InfoGAN (5) which does not have the classification agent, and **A**uxiliary **C**lassifier **GAN** (AC-GAN) (22) which only performs classification, our propsoed DA-GAN performs face verification with an intrigued data augmentation. DA-GAN is a novel and practical model for efficient data augmentation and it is really effective in practice as proved in Sec. 4. DA-GAN generates the data in a completely different way from InfoGAN (5) and AC-GAN (22) which generate images from a random noise input or abstract semantic labels. Therefore, inferior to our model, those existing GAN-like models cannot exploit useful and rich prior information (*e.g.*, the shape, pose of faces) for effective data generation and augmentation. They cannot fully control the generated images. In contrast, DA-GAN can fully control the generated images and adjust the face pose (*e.g.*, yaw angles) distribution which is extremely unbalanced in real-world scenarios. DA-GAN can facilitate training more accurate face analysis models to solve the large pose variation problem and other relevant problems in unconstrained face recognition.

Our proposed DA-GAN shares a similar idea with TP-GAN (13) that considers face synthesis based on GAN framework, and Apple GAN (28) that considers learning from simulated and unsupervised images through adversarial training. Our method differs from them in following aspects: 1) DA-GAN aims to synthesize photorealistic and identity preserving profile faces to address the large variance issue in unconstrained face recognition, whereas TP-GAN (13) tries to recover a frontal face from a profile view and Apple GAN (28) is designed for much simpler scenarios (*e.g.*, eye and hand image refinement); 2) TP-GAN (13) and Apple GAN (28) suffer from categorical information loss which limits their effectiveness in promoting recognition performance. In contrast, our proposed DA-GAN architecture effectively overcomes this issue by introducing dual discriminator agents.

## 3 Dual-Agent GAN

### 3.1 Simulator

The main challenge for unconstrained face recognition lies in the large variation and few profile face images for each subject, which is the main obstacle for learning a well-performed pose-invariant model. To address this problem, we simulate face images with various pre-defined poses (*i.e.*, yaw angles), which explicitly augments the available training data without extra human annotation efforts and balances the pose distribution.

In particular, as shown in Figure. 2, we first extracts the face **R**egion **o**f **I**nterest (RoI) from each available real face image, and estimate 68 facial landmark points using the **R**ecurrent **A**ttentive-**R**efinement (RAR) framework (31), which is robust to illumination changes and does not require a shape model in advance. We then estimate a transformation matrix between the detected 2D landmarks and the corresponding landmarks in the **3D M**orphable **M**odel (3D MM) using least-squares fit (35). Finally, we simulate profile face images in various poses with pre-defined yaw angles.

However, the performance of the simulator decreases dramatically under large poses (*e.g.*, yaw angles $\in \{[-90°, -60°] \cup [+60°, +90°]\}$) due to artifacts and severe texture losses, misleading the network to overfit to fake information only presented in synthetic images and fail to generalize well on real data.

### 3.2 Generator

In order to generate photorealistic and identity preserving profile view face images which are truely benefical for unconstrained face recognition, we further refine the above-mentioned simulated profile face images with the proposed DA-GAN.

Inspired by the recent success of FCN-based methods on image-to-image applications (17; 9) and the leading performance of skip-net on recognition tasks (12; 33), we modify a skip-net (ResNet (12)) into a FCN-based architecture as the generator $G_\theta : \mathbb{R}^{H \times W \times C} \mapsto \mathbb{R}^{H \times W \times C}$ of DA-GAN to learn a highly non-linear transformation for profile face image refinement, where $\theta$ are the network parameters for the generator, and $H$, $W$, and $C$ denote the image height, width, and channel number, repetively.

Contextual information from global and local regions compensates each other and naturally benefits face recognition. The hierarchical features within a skip-net are multi-scale in nature due to the increasing receptive field sizes, which are combined together via skip connections. Such a combined representation comprehensively maintains the contextual information, which is crucial for

artifact removal, fragement stitching, and texture padding. Moreover, the FCN-based architecture is advantageous for generating high-resolution image-level results. More details are provided in Sec. 4.

More formally, let the simulated profile face image be denoted by $x$ and the refined face image be denoted by $\tilde{x}$, then

$$\tilde{x} := G_\theta(x). \tag{1}$$

The key requirements for DA-GAN are that the refined face image $\tilde{x}$ should look like a real face image in appearance while preserving the intrinsic identity and pose information from the simulator.

To this end, we propose to learn $\theta$ by minimizing a combination of three losses:

$$\mathcal{L}_{G_\theta} = (-\mathcal{L}_{adv} + \lambda_1 \mathcal{L}_{ip}) + \lambda_2 \mathcal{L}_{pp}, \tag{2}$$

where $\mathcal{L}_{adv}$ is the **adv**ersarial loss for adding realism to the synthetic images and alleviating artifacts, $\mathcal{L}_{ip}$ is the **i**dentity **p**erception loss for preserving the identity information, and $\mathcal{L}_{pp}$ is the **p**ose **p**erception loss for preserving pose and texture information.

$\mathcal{L}_{pp}$ is a pixel-wise $\ell_1$ loss, which is introduced to enforce the pose (*i.e.*, yaw angle) consistency for the synthetic profile face images before and after the refinement via DA-GAN:

$$\mathcal{L}_{pp} = \frac{1}{W \times H} \sum_{i}^{W} \sum_{j}^{H} |x_{i,j} - \tilde{x}_{i,j}|, \tag{3}$$

where $i, j$ traverse all pixels of $x$ and $\tilde{x}$.

Although $\mathcal{L}_{pp}$ may lead some over smooth effects to the refined results, it is still an essential part for both pose and texture information preserving and accelerated optimization.

To add realism to the synthetic images to really benefit face recognition performance, we need to narrow the gap between the distributions of synthetic and real images. An ideal generator will make it impossible to classify a given image as real or refined with high confidence. Meanwhile, preserving the identity information is the essential and critical part for recognition. An ideal generator will generate the refined face images that have small intra-class distance and large inter-class distance in the feature space spanned by the deep neural networks for unconstrained face recognition. These motivate the use of an adversarial pixel-wise discriminator with dual agents.

### 3.3 Dual-agent discriminator

To incorporate the prior knowledge from the profile faces' distribution and domain knowledge of identities' distribution, we herein introduce a discriminator with dual agents for distinguishing real *v.s.* fake and identities simultaneously. To facilitate this process, we leverage an auto-encoder as the discriminator $D_\phi : \mathbb{R}^{H \times W \times C} \mapsto \mathbb{R}^{H \times W \times C}$ to be as simple as possible to avoid typical GAN tricks, which first projects the input real / fake face image into high-dimensional feature space through several **Conv**olution (Conv) and **F**ully **C**onnected (FC) layers of the encoder and then transformed back to the image-level representation through several **Deconv**olution (Deconv) and Conv layers of the decoder, as shown in Figure. 2. $\phi$ are the network parameters for the discriminator. More details are provided in Sec. 4.

One agent of $D_\phi$ is trained with $\mathcal{L}_{adv}$ to minimize the Wasserstein distance with a boundary equilibrium regularization term for maintaining a balance between the generator and discriminator losses as first introduced in (2),

$$\mathcal{L}_{adv} = \sum_{j} |y_j - D_\phi(y_j)| - k_t \sum_{i} |\tilde{x}_i - D_\phi(\tilde{x}_i)|, \tag{4}$$

where $y$ denotes the real face image, $k_t$ is a boundary equilibrium regularization term using Proportional Control Theory to maintain the equilibrium $\mathbb{E}[\sum_i |\tilde{x}_i - D_\phi(\tilde{x}_i)|] = \gamma \mathbb{E}[\sum_j |y_j - D_\phi(y_j)|]$, $\gamma$ is the diversity ratio.

Here $k_t$ is updated by

$$k_{t+1} = k_t + \alpha(\gamma \sum_{j} |y_j - D_\phi(y_j)| - \sum_{i} |\tilde{x}_i - D_\phi(\tilde{x}_i)|), \tag{5}$$

where $\alpha$ is the learning rate (proportional gain) for $k$. In essence, Eq.(5) can be thought of as a form of close-loop feedback control in which $k_t$ is adjusted at each step.

$\mathcal{L}_{\text{adv}}$ serves as a supervision to push the refined face image to reside in the manifold of real images. It can prevent the blurry effect, alleviate artifacts and produce visually pleasing results.

The other agent of $D_\phi$ is trained with $\mathcal{L}_{\text{ip}}$ to preserve the identity discriminability of the refined face images. Specially, we define $\mathcal{L}_{\text{ip}}$ with the multi-class cross-entropy loss based on the output from the bottleneck layer of $D_\phi$.

$$\mathcal{L}_{\text{ip}} = \frac{1}{N} \sum_j -(Y_j log(D_\phi(y_j)) + (1 - Y_j)log(1 - D_\phi(y_j)))$$
$$+ \frac{1}{N} \sum_i -(Y_i log(D_\phi(\tilde{x}_i)) + (1 - Y_i)log(1 - D_\phi(\tilde{x}_i))), \tag{6}$$

where $Y$ is the identity ground truth.

Thus, minimizing $\mathcal{L}_{\text{ip}}$ would encourage deep features of the refined face images belonging to the same identity to be close to each other. If one visualizes the learned deep features in the high-dimensional space, the learned deep features of refined face image set form several compact clusters and each cluster may be far away from others. Each cluster has a small variance. In this way, the refined face images are enforced with well preserved identity information. We also conduct experiments for illustration.

Using $\mathcal{L}_{\text{ip}}$ alone makes the results prone to annoying artifacts, because the search for a local minimum of $\mathcal{L}_{\text{ip}}$ may go through a path that resides outside the manifold of natural face images. Thus, we combine $\mathcal{L}_{\text{ip}}$ with $\mathcal{L}_{\text{adv}}$ as the final objective function for $D_\phi$ to ensure that the search resides in that manifold and produces photorealistic and identity preserving face image:

$$\mathcal{L}_{D_\phi} = \mathcal{L}_{\text{adv}} + \lambda_1 \mathcal{L}_{\text{ip}}. \tag{7}$$

### 3.4 Loss functions for training

The goal of DA-GAN is to use a set of unlabeled real face images $y$ to learn a generator $G_\theta$ that adaptively refines a simulated profile face image $x$. The overall objective function for DA-GAN is:

$$\begin{cases} \mathcal{L}_{D_\phi} = \mathcal{L}_{\text{adv}} + \lambda_1 \mathcal{L}_{\text{ip}}, \\ \mathcal{L}_{G_\theta} = (-\mathcal{L}_{\text{adv}} + \lambda_1 \mathcal{L}_{\text{ip}}) + \lambda_2 \mathcal{L}_{\text{pp}}. \end{cases} \tag{8}$$

We optimize DA-GAN by alternatively optimizing $D_\phi$ and $G_\theta$ for each training iteration. Similar as in (2), we measure the convergence of DA-GAN by using the boundary equilibrium concept: we can frame the convergence process as finding the closest reconstruction $\sum_j |y_j - D_\phi(y_j)|$ with the lowest absolute value of the instantaneous process error for the Proportion Control Theory $|\gamma \sum_j |y_j - D_\phi(y_j)| - \sum_i |\tilde{x}_i - D_\phi(\tilde{x}_i)||$. This measurement can be formulated as:

$$\mathcal{L}_{\text{con}} = \sum_j |y_j - D_\phi(y_j)| + |\gamma \sum_j |y_j - D_\phi(y_j)| - \sum_i |\tilde{x}_i - D_\phi(\tilde{x}_i)||. \tag{9}$$

$\mathcal{L}_{\text{con}}$ can be used to determine when the network has reached its final state or if the model has collapsed. Detailed algorithm on the training procedures is provided in supplementary material Sec. 1.

## 4 Experiments

### 4.1 Experimental settings

**Benchmark dataset:** Except for synthesizing natural looking profile view face images, the proposed DA-GAN also aims to generate identity preserving face images for accurate face-centric analysis with state-of-the-art deep learning models. Therefore, we evaluate the possibility of "recognition via generation" of DA-GAN on the most challenging unconstrained face recognition benchmark dataset IJB-A (15). IJB-A (15) contains both images and video frames from 500 subjects with 5,397 images and 2,042 videos that are split into 20,412 frames, 11.4 images and 4.2 videos per subject, captured from in-the-wild environment to avoid the near frontal bias, along with protocols for evaluation of both *verification* (1:1 comparison) and *identification* (1:$N$ search) tasks. For training and testing, 10 random splits are provided by each protocol, respectively. More details are provided in supplementary material Sec. 2.

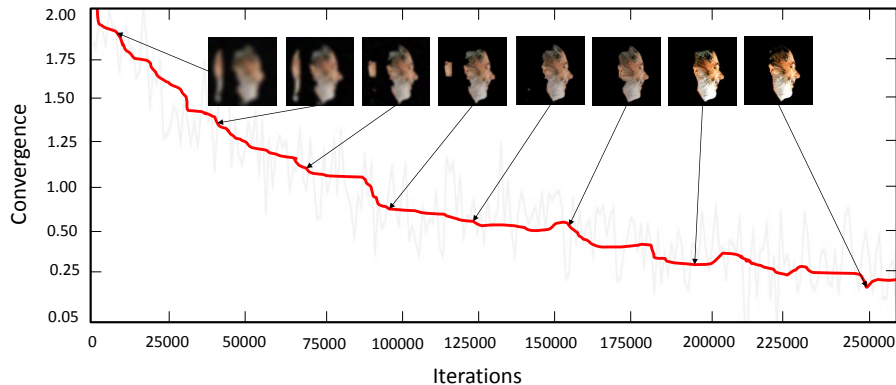

Figure 3: Quality of refined results *w.r.t.* the network convergence measurement $\mathcal{L}_{\text{con}}$.

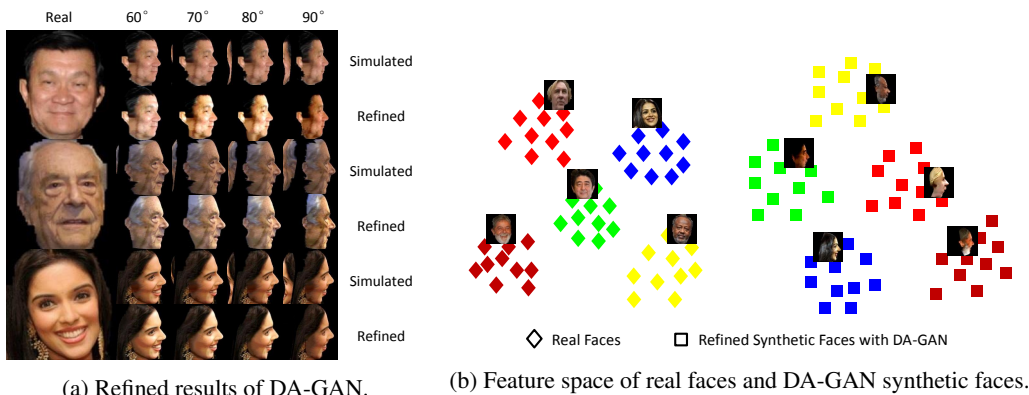

(a) Refined results of DA-GAN.

(b) Feature space of real faces and DA-GAN synthetic faces.

Figure 4: Qualitative analysis of DA-GAN.

**Reproducibility:** The proposed method is implemented by extending the Keras framework (6). All networks are trained on three NVIDIA GeForce GTX TITAN X GPUs with 12GB memory for each. Please refer to supplementary material Sec. 3 & 4 for full details on network architectures and training procedures.

## 4.2 Results and discussions

**Qualitative results – DA-GAN:** In order to illustrate the compelling perceptual results generated by the proposed DA-GAN, we first visualize the quality of refined results *w.r.t.* the network convergence measurement $\mathcal{L}_{\text{con}}$, as shown in Figure. 3. As can be seen, our DA-GAN ensures a fast yet stable convergence through the carefully designed optimization scheme and boundary equilibrium regularization term. The network convergence measurement $\mathcal{L}_{\text{con}}$ correlates well with image fidelity.

Most of the previous works (31; 32; 35) on profile view synthesis are dedicated to address this problem within a pose range of $\pm 60°$. Because it is commonly believed that with a pose that is larger than $60°$, it is difficult for a model to generate faithful profile view images. Similarly, our simulator is also good at normalizing small posed faces while suffers severe artifacts and texture losses under large poses (*e.g.*, yaw angles $\in \{[-90°, -60°] \cup [+60°, +90°]\}$), as shown in Figure. 4a the first row for each subject. However, with enough training data and proper architecture and objective function design of the proposed DA-GAN, it is in fact feasible to further refine such synthetic profile face images under very large poses for high-quality natural looking results generation, as shown in Figure. 4a the second row for each subject. Compared with the raw simulated faces, the refined results by DA-GAN present a good photorealistic quality. More visualized samples are provided in supplementary material Sec. 5.

To verify the superiority of DA-GAN as well as the contribution of each component, we also compare the qualitative results produced by the vanilla GAN (10), Apple GAN (28), BE-GAN (2) and three variations of DA-GAN in terms of w/o $\mathcal{L}_{\text{adv}}$, $\mathcal{L}_{\text{ip}}$, $\mathcal{L}_{\text{pp}}$ in each case, repectively. Please refer to supplementary material Sec. 5 for details.

Table 1: Performance comparison of DA-GAN with state-of-the-arts on IJB-A (15) verification protocol. For all metrics, a higher number means better performance. The results are averaged over 10 testing splits. Symbol "-" implies that the result is not reported for that method. Standard deviation is not available for some methods. The results offered by our proposed method are highlighted in bold.

| Method | Face verification | | |
|---|---|---|---|
| | TAR @ FAR=0.10 | TAR @ FAR=0.01 | TAR @ FAR=0.001 |
| OpenBR (15) | $0.433 \pm 0.006$ | $0.236 \pm 0.009$ | $0.104 \pm 0.014$ |
| GOTS (15) | $0.627 \pm 0.012$ | $0.406 \pm 0.014$ | $0.198 \pm 0.008$ |
| Pooling faces (11) | 0.631 | 0.309 | - |
| LSFS (30) | $0.895 \pm 0.013$ | $0.733 \pm 0.034$ | $0.514 \pm 0.060$ |
| Deep Multi-pose (1) | 0.911 | 0.787 | - |
| $DCNN_{manual}$ (4) | $0.947 \pm 0.011$ | $0.787 \pm 0.043$ | - |
| Triplet Similarity (27) | $0.945 \pm 0.002$ | $0.790 \pm 0.030$ | $0.590 \pm 0.050$ |
| VGG-Face (23) | - | $0.805 \pm 0.030$ | - |
| PAMs (19) | $0.652 \pm 0.037$ | $0.826 \pm 0.018$ | - |
| $DCNN_{fusion}$ (3) | $0.967 \pm 0.009$ | $0.838 \pm 0.042$ | - |
| Masi *et al.* (20) | - | 0.886 | 0.725 |
| Triplet Embedding (27) | $0.964 \pm 0.005$ | $0.900 \pm 0.010$ | $0.813 \pm 0.020$ |
| All-In-One (25) | $0.976 \pm 0.004$ | $0.922 \pm 0.010$ | $0.823 \pm 0.020$ |
| Template Adaptation (8) | $0.979 \pm 0.004$ | $0.939 \pm 0.013$ | $0.836 \pm 0.027$ |
| NAN (34) | $0.978 \pm 0.003$ | $0.941 \pm 0.008$ | $0.881 \pm 0.011$ |
| $\ell_2$-softmax (24) | $0.984 \pm 0.002$ | $0.970 \pm 0.004$ | $0.943 \pm 0.005$ |
| b1 | $0.989 \pm 0.003$ | $0.963 \pm 0.007$ | $0.920 \pm 0.006$ |
| b2 | $0.978 \pm 0.003$ | $0.950 \pm 0.009$ | $0.901 \pm 0.008$ |
| DA-GAN (ours) | $\mathbf{0.991 \pm 0.003}$ | $\mathbf{0.976 \pm 0.007}$ | $\mathbf{0.930 \pm 0.005}$ |

To gain insights into the effectivenss of identity preserving quality of our DA-GAN, we further use t-SNE (18) to visualize the deep features of both refined profile faces and real faces in a 2D space in Figure. 4b. As can be seen, the refined profile face images present small intra-class distance and large inter-class distance, which is similar to those of real faces. This reveals that DA-GAN ensures well preserved identity information with the auxiliary agent for $\mathcal{L}_{ip}$.

**Quantitative results – "recognition via generation":** To quantitatively verify the superiority of "recognition via generation" of DA-GAN, we conduct unconstrained face recognition (*i.e.*, verification and identification) on IJB-A (15) benchmark dataset with three different settings. In the three settings, the pre-trained deep recognition models are respectively fine-tuned on the original training data of each split without extra data (baseline 1: b1), the original training data of each split with extra synthetic faces by our simulator (baseline 2: b2), and the original training data of each split with extra refined faces by our DA-GAN (our method: "recognition via generation" framework based on DA-GAN, DA-GAN for short). The performance comparison of DA-GAN with the two baselines and other state-of-the-arts on IJB-A (15) unconstrained face verification and identification protocols are given in Table. 1 and Table. 2.

We can observe that even with extra training data, b2 presents inferior performance than b1 for all metrics of both face verification and identification. This demonstrates that naively learning from synthetic images can be problematic due to a gap between synthetic and real image distributions – synthetic data is often not realistic enough with artifacts and severe texture losses, misleading the network to overfit to fake information only presented in synthetic images and fail to generalize well on real data. In contrast, with the injection of photorealistic and identity preserving faces generated by DA-GAN without extra human annotation efforts, our method outperforms b1 by $1.00\%$ for TAR @ FAR=0.001 of verification and $1.50\%$ for FNIR @ FPIR=0.01, $0.50\%$ for Rank-1 of identification. Our method achieves comparable performance with $\ell_2$-softmax (24), which employ a much more computational complex recognition model even without fine-tuning or template adaptation procedures as we do. Moreover, DA-GAN outperforms NAN (34) by $4.90\%$ for TAR @ FAR=0.001 of verification and $7.30\%$ for FNIR @ FPIR=0.01, $1.30\%$ for Rank1 of identification. These results won the $1^{st}$ places on verification and identification tracks in NIST IJB-A 2017 face recognition competitions[3]. This well verified the promising potential of synthetic face images by our DA-GAN on the large-scale and challenging unconstrained face recognition problem.

Table 2: Performance comparison of DA-GAN with state-of-the-arts on IJB-A (15) identification protocol. For FNIR metric, a lower number means better performance. For the other metrics, a higher number means better performance. The results offered by our proposed method are highlighted in bold.

| Method | Face identification | | | |
|---|---|---|---|---|
| | FNIR @ FPIR=0.10 | FNIR @ FPIR=0.01 | Rank1 | Rank5 |
| OpenBR (15) | $0.851 \pm 0.028$ | $0.934 \pm 0.017$ | $0.246 \pm 0.011$ | $0.375 \pm 0.008$ |
| GOTS (15) | $0.765 \pm 0.033$ | $0.953 \pm 0.024$ | $0.433 \pm 0.021$ | $0.595 \pm 0.020$ |
| B-CNN (7) | $0.659 \pm 0.032$ | $0.857 \pm 0.027$ | $0.588 \pm 0.020$ | $0.796 \pm 0.017$ |
| LSFS (30) | $0.387 \pm 0.032$ | $0.617 \pm 0.063$ | $0.820 \pm 0.024$ | $0.929 \pm 0.013$ |
| Pooling faces (11) | - | - | 0.846 | 0.933 |
| Deep Multi-pose (1) | 0.250 | 0.480 | 0.846 | 0.927 |
| DCNN$_{manual}$ (4) | - | - | $0.852 \pm 0.018$ | $0.937 \pm 0.010$ |
| Triplet Similarity (27) | $0.246 \pm 0.014$ | $0.444 \pm 0.065$ | $0.880 \pm 0.015$ | $0.950 \pm 0.007$ |
| VGG-Face (23) | $0.33 \pm 0.031$ | $0.539 \pm 0.077$ | $0.913 \pm 0.011$ | - |
| PAMs (19) | - | - | $0.840 \pm 0.012$ | $0.925 \pm 0.008$ |
| DCNN$_{fusion}$ (3) | $0.210 \pm 0.033$ | $0.423 \pm 0.094$ | $0.903 \pm 0.012$ | $0.965 \pm 0.008$ |
| Masi *et al.* (20) | - | - | 0.906 | 0.962 |
| Triplet Embedding (27) | $0.137 \pm 0.014$ | $0.247 \pm 0.030$ | $0.932 \pm 0.010$ | - |
| Template Adaptation (8) | $0.118 \pm 0.016$ | $0.226 \pm 0.049$ | $0.928 \pm 0.010$ | $0.977 \pm 0.004$ |
| All-In-One (25) | $0.113 \pm 0.014$ | $0.208 \pm 0.020$ | $0.947 \pm 0.008$ | - |
| NAN (34) | $0.083 \pm 0.009$ | $0.183 \pm 0.041$ | $0.958 \pm 0.005$ | $0.980 \pm 0.005$ |
| $\ell_2$-softmax (24) | $0.044 \pm 0.006$ | $0.085 \pm 0.041$ | $0.973 \pm 0.005$ | - |
| b1 | $0.068 \pm 0.010$ | $0.125 \pm 0.035$ | $0.966 \pm 0.006$ | $0.987 \pm 0.003$ |
| b2 | $0.108 \pm 0.008$ | $0.179 \pm 0.042$ | $0.960 \pm 0.007$ | $0.982 \pm 0.004$ |
| DA-GAN (ours) | $\mathbf{0.051 \pm 0.009}$ | $\mathbf{0.110 \pm 0.039}$ | $\mathbf{0.971 \pm 0.007}$ | $\mathbf{0.989 \pm 0.003}$ |

Finally, we visualize the verification and identification closed set results for IJB-A (15) split1 to gain insights into unconstrained face recognition with the proposed "recognition via generation" framework based on DA-GAN. For fully detailed visualization results in high resolution and corresponding analysis, please refer to supplementary material Sec. 6 & 7.

# 5 Conclusion

We proposed a novel **D**ual-**A**gent **G**enerative **A**dversarial **N**etwork (DA-GAN) for photorealistic and identity preserving profile face synthesis. DA-GAN combines prior knowledge from data distribution (adversarial training) and domain knowledge of faces (pose and identity perception loss) to exactly recover the lost information inherent in projecting a 3D face into the 2D image space. DA-GAN can be optimized in a fast yet stable way with an imposed boundary equilibrium regularization term that balances the power of the discriminator against the generator. One promising potential application of the proposed DA-GAN is for solving generic transfer learning problems more effectively. Qualitative and quantitative experiments verify the possibility of our "recognition via generation" framework, which achieved the top performance on the large-scale and challenging NIST IJB-A unconstrained face recognition benchmark without extra human annotation efforts. Based on DA-GAN, we won the 1st places on verification and identification tracks in NIST IJB-A 2017 face recognition competitions. It would be interesting to apply DA-GAN for other transfer learning applications in the future.

## Acknowledgement

The work of Jian Zhao was partially supported by **C**hina **S**cholarship **C**ouncil (CSC) grant 201503170248.

The work of Jiashi Feng was partially supported by National University of Singapore startup grant R-263-000-C08-133, Ministry of Education of Singapore AcRF Tier One grant R-263-000-C21-112 and NUS IDS grant R-263-000-C67-646.

We would like to thank Junliang Xing (Institute of Automation, Chinese Academy of Sciences), Hengzhu Liu, and Xucan Chen (National University of Defense Technology) for helpful discussions.

performance on both tracks on 26th, Apirl, 2017. The IJB-A benchmark dataset, relevant information and leaderboard can be found at https://www.nist.gov/programs-projects/face-challenges.

## Footnotes

*Homepage: https://zhaoj9014.github.io/.

†Jian Zhao and Zhecan Wang were interns at Panasonic R&D Center Singapore during this work.

[3]We submitted our results for both verification and identification protocols to NIST IJB-A 2017 face recognition competition committee on 29th, March, 2017. We received the official notification on our top

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
