[Supplementary Material]

# Supplementary Material for
# Dual-Agent GANs for Photorealistic and Identity Preserving Profile Face Synthesis

**Jian Zhao**[1,2*†]    **Lin Xiong**[3]    **Karlekar Jayashree**[3]    **Jianshu Li**[1]    **Fang Zhao**[1]
**Zhecan Wang**[4†]    **Sugiri Pranata**[3]    **Shengmei Shen**[3]
**Shuicheng Yan**[1,5]    **Jiashi Feng**[1]
[1]National University of Singapore    [2]National University of Defense Technology
[3] Panasonic R&D Center Singapore    [4] Franklin. W. Olin College of Engineering
[5] Qihoo 360 AI Institute
{zhaojian90, jianshu}@u.nus.edu    {lin.xiong, karlekar.jayashree, sugiri.pranata, shengmei.shen}@sg.panasonic.com
zhecan.wang@students.olin.edu    {elezhf, eleyans, elefjia}@u.nus.edu

## Abstract

In this supplementary material, we present fully detailed information on 1) learning algorithm of the proposed **D**ual-**A**gent **G**enerative **A**dversarial **N**etwork (DA-GAN) model; 2) details on the IJB-A benchmark dataset (7); 3) network architectures; 4) training details; 5) qualitative analysis of DA-GAN; 6) high-resolution visualized verification results for IJB-A (7) split1; 7) high-resolution visualized identification results for IJB-A (7) split1.

## 1   Learning algorithm of DA-GAN model

We summarize detailed the training procedures of our DA-GAN in Algorithm. 1.

## 2   Details on the IJB-A benchmark dataset

IJB-A  (7) contains both images and video frames from 500 subjects with 5,397 images and 2,042 videos that are split into 20,412 frames, 11.4 images and 4.2 videos per subject, captured from in-the-wild environment to avoid the near frontal bias, along with protocols for evaluation of both *verification* (1:1 comparison) and *identification* (1:$N$ search) tasks.  For training and testing, 10 random splits are provided by each protocol, respectively.

IJB-A (7) defines the minimal facial representation unit to be a "template" enrolled with multiple face images and / or video frames under extreme conditions of pose, expression, occlusion, and illumination. Such problem setting is aligned better with real-world scenario where each subject's appearance is more likely to be captured more than once using different approaches, turning the traditional face recognition problem into a more challenging set-to-set matching problem under extreme conditions in the wild. The verification task requires the evaluation system to determine whether two input face templates are of the same subject or not. At a given threshold, the **R**eceiver **O**perating **C**haracteristic (ROC) analysis measures the **T**rue **A**ccept **R**ate (TAR), which is the fraction of genuine comparisons that correctly exceed the threshold, and the **F**alse **A**ccept **R**ate (FAR), which is the fraction of impostor comparisons that incorrectly exceed the threshold. For identification, the evaluation system needs to determine the subject matching a probe identity from a closed set or an open set. For a closed set, the **C**umulative **M**atch **C**haracteristic (CMC) analysis measures the percentage of probe searches returning probe gallery mates within a given Rank. For an open set,

**Algorithm 1** Learning algorithm of DA-GAN
---
**Input:** Sets of synthetic profile face images $x_i$, real face images $y_j$, and the associated identity
   labels $Y_i$, max number of epoches (nb_e), batch size (b), number of network updates per step
   (nb_s), input size (im_w, im_h, im_c), weight decay, learning rate (lr), $k_0$, $\lambda_1$, $\lambda_2$, $\alpha$, $\gamma$;
**Output:** DA-GAN generator $G_\theta$ and discriminator $D_\phi$;
   **for** e=1, $\cdots$ , nb_e **do**
      **for** s=1, $\cdots$ , nb_s **do**
         1. Optimize $D_\phi$;
         2. Optimize $G_\theta$;
         3. Update $k_t$;
         4. Measure network convergence $\mathcal{L}_{\mathrm{con}}$;
         5. Visualize intermediate results;
      **end for**
      Archive $G_\theta$ and $D_\phi$ models for each training epoch;
   **end for**
---

Figure 1: Framework overview of "recognition via generation". We transfer learn two state-of-the-art deep neural networks – ResNext-50 (11) and GoogleNet-BN (9) from source domain to target domain extended by DA-GAN. We ensemble the compensate two-view information from the two models to train template adapted SVMs (2). The resulted margins are robust and discriminative for unconstrained face recognition. Best viewed in color.

at a given threshold, the evaluation system measures the **F**alse **P**ositive **I**dentification **R**ate (FPIR), which is the fraction of comparisons between probe templates and non-mate gallery templates that corresponds to a match score exceeding the threshold, and the **F**alse **N**egative **I**dentification **R**ate (FNIR), which is the fraction of probe searches that fail to match a mated gallery template above a score of the threshold. More details on the evaluation metrics can be found in (7).

## 3 Network architectures

- Simulator: RAR framework (10) (face RoI extraction & 68 facial landmark detection), 3D MM (12) (profile face image simulation with pre-defined yaw angles).

- Generator: Input $224 \times 224 \times 3$, Conv $64 \times 7 \times 7$, ReLU[3], BN[4], 10×Residual block (Conv $64 \times 7 \times 7$, ReLU, BN, Conv $64 \times 7 \times 7$, Ele-Sum[5], ReLU, BN), Conv $3 \times 1 \times 1$.

- Discriminator: Input $224 \times 224 \times 3$, Conv $3 \times 3 \times 3$, ReLU, Transition down (Conv $128 \times 3 \times 3$, ReLU, Conv $128 \times 3 \times 3/2$, ReLU, Conv $256 \times 3 \times 3$, ReLU, Conv $256 \times 3 \times 3/2$, ReLU, Conv $384 \times 3 \times 3$, ReLU, Conv $384 \times 3 \times 3/2$, ReLU), Flatten, FC 784, Reshape, Transition up (Conv $128 \times 3 \times 3$, ReLU, Deconv $128 \times 3 \times 3/2$, ReLU, Conv $128 \times 3 \times 3$, ReLU, Deconv $128 \times 3 \times 3/2$, ReLU, Conv $128 \times 3 \times 3$, ReLU, Deconv $128 \times 3 \times 3/2$, ReLU), Conv $3 \times 1 \times 1$, ReLU.

- Deep recognition models: Input $224 \times 224 \times 3$, ResNext-50 (cardinality = 32) (11) & GoogleNet-BN (9) (model fusion), template adapted **S**upport **V**ector **M**achine (SVM) (2) (metric learning).

The overview of our proposed "recognition via generation" framework is illustrated in Figure. 1. We transfer learn two state-of-the-art deep neural networks – ResNext-50 (11) and GoogleNet-BN (9) from source domain (MS-Celeb-1M (4), removed overlapping parts with IJB-A (7)) to target domain of IJB-A (7) extended by DA-GAN. We ensemble the compensate two-view information (learned deep features) from the ResNext-50 (11) and GoogleNet-BN (9) models to train template adapted SVMs (2). The resulted margins are robust and discriminative for unconstrained face recognition.

## 4 Training details

- DA-GAN: 1) Extract face RoIs from the available training data of each IJB-A (7) split, and detect 68 facial landmark points using the RAR framework (10). 2) Simulate profile faces with pre-defined yaw angles $\in \{\pm 10, \pm 20, \pm 30, \pm 40, \pm 50, \pm 60, \pm 70, \pm 80, \pm 90\}$ using 3D MM (12). 3) Train DA-GAN using Adam with mini-batch (FC 333 with Softmax appended to the output of the bottleneck layer of $D_\phi$ for $\mathcal{L}_{\text{ip}}$ during training); set the mini-batch size to 16; $W = 224$, $H = 224$, $C = 3$; initialize DA-GAN using vanishing residuals; set an initial learning rate to $5 \times 10^{-5}$, decaying by a factor of 2 when $\mathcal{L}_{\text{con}}$ stalls; set the weight decay to $5 \times 10^{-4}$; set $k_0 = 0$; $\lambda_1 = 2.5 \times 10^{-2}$, $\lambda_2 = 3 \times 10^{-2}$, $\alpha = 1 \times 10^{-3}$, $\gamma = 5 \times 10^{-1}$; alternatively optimize discriminator $D_\phi$, generator $G_\theta$ and update $k_t$ for each mini-batch.

- Deep recognition models: 1) Set the mini-batch size to 256; $W = 224$, $H = 224$, $C = 3$; set an initial learning rate to 0.01 and divided by 10 every 30 epochs; set the weight decay to $1 \times 10^{-4}$; set the momentum to 0.9. 2) Pre-process MS-Celeb-1M (4) data, including overlapping part removal with IJB-A (7) and face RoI extraction, resulting in 4,356,052 face images for 53,317 subjects in total. 3) Train ResNext-50 (cardinality = 32) (11) & GoogleNet-BN (9) using **S**tochastic **G**radient **D**escent (SGD) on the cleaned MS-Celeb-1M (4) data. 4) Reset the learning rate to 0.0001 and divided by 10 every 10 epoches. 5) Inject the refined profile face images and video frames into IJB-A (7) each split training data and fine-tune the pre-trained deep recognition models.

- Template adapted SVM models: 1) Concat the learned pose-invariant features from the penultimate layers of deep recognition models ($\mathbb{R}^{2048}$ C-Sum$^6$ $\mathbb{R}^{1024} \mapsto \mathbb{R}^{3072}$). 2) Train template adapted SVM models similarly as introduced in (2).

More formally, the template adapted SVMs are learned by optimizing the following $\ell_2$-regularized objective function:

$$\mathcal{L}_{\text{SVM}} = \min_{w} \frac{1}{2} w^T w + \lambda_+ \sum_{i=1}^{N_+} \max \left[ 0, 1 - y_i w^T f_F \left( \mathbf{x}_i \right) \right]^2 + \lambda_- \sum_{j=1}^{N_-} \max \left[ 0, 1 - y_j w^T f_F \left( \mathbf{x}_j \right) \right]^2, \quad (1)$$

where $f_F(\cdot)$ denotes the non-linear function learned by our deep recognition models, $x$ denote the face media, $w$ denote the weights including bias term, $y_i \in \{-1, 1\}$ denotes the label indicating whether the current sample being negative or possible, $N_+$ indicates the number of positive samples, $N_-$ indicates the number of negative ones, $N_- \gg N_+$, the constraint for negative samples $\lambda_- = C \frac{N_+ + N_-}{2N_-}$, the constraint for positive samples $\lambda_+ = C \frac{N_+ + N_-}{2N_+}$, $C$ is a trade-off factor, and we set it to 20 in our method.

---

$^6$C-Sum is short for concat.

Since a template contains both face images and / or video frames, containing large variances in terms of media modality, pose, expression, occlusion, and illumination. In order to better address the underlying distracting factors within each template, we split each template into several sub-templates according to the prior information on the media source (*e.g.*, image / video). In particular, for the deep features from a video sequence, we perform mean encoding to generate the corresponding representation.

Let $t_j^V$ be the mean encoding of the $j^{th}$ video sequence, then

$$t_j^V = \frac{1}{N_j^V} \sum_{i=1}^{N_j^V} f_F(\mathbf{x}_i),\qquad(2)$$

where $N_j^V$ is the number of frame in the $j^{th}$ video sequence, $\mathbf{x}_i$ denotes the $i^{th}$ frame of video $j$.

Thus, the representations for the $a^{th}$ template can be expressed as

$$T_a = \left\{ t_i^I, ..., t_{N_a}^V \right\},\qquad(3)$$

where $t_i^I$ denotes the sub-template for the $i^{th}$ image, $t_{N_a}^V$ denotes the sub-template for the $N_a^{th}$ video.

The media-level deep features are further $L_2$-normalized for training template adapted SVMs (2). For verification, the positive sample of template specific SVM is a probe template, and the large-scale negative samples consist of the whole training set. For identification, the probe template specific SVMs adopt the whole training set as the large-scale negative samples; whereas for gallery template specific SVM, other gallery templates and the whole training set are bundled together as the large-scale negative samples.

Based on one shot similarity, we compute the fine-grained similarity between two sub-template representations $p$ and $q$ via $s(p,q) = \frac{1}{2}\mathcal{P}(q) + \frac{1}{2}\mathcal{Q}(p)$, where $\mathcal{P}(\cdot)$ denotes the trained probe template specific SVM model and $\mathcal{Q}(\cdot)$ indicates the trained gallery template specific SVM model.

As described in Eq. (3), a template may contain various number of sub-templates. Thus, finally we merge the resulting multiple matching scores into a single measurement to determine the face identity for each template pair,

$$s(T_a, T_b) = \frac{\sum\limits_{t_i \in T_a, t_j \in T_b} s(t_i, t_j) e^{\beta \, s(t_i, t_j)}}{\sum\limits_{t_i \in T_a, t_j \in T_b} e^{\beta \, s(t_i, t_j)}},\qquad(4)$$

where $\beta$ is a bandwidth factor, and we set it to 0 in our method.

## 5  Qualitative analysis of DA-GAN

We visualize the high-resolution refined results of DA-GAN under various poses with yaw angles ranging from $-90°$ to $-10°$ and $+10°$ to $+90°$ at a stride of $10°$ in Figure. 2 and Figure. 3 to verify the compelling perceptual quality of DA-GAN. As can be seen, DA-GAN is able to adaptively remove artifacts (*e.g.*, face fragments and black holes) introduced by the simulator, stitch fragments, and compensate texture losses in terms of facial details and color realism, especially for large poses. As a result, the refined faces of DA-GAN present more intuitively photorealistic and natural characteristics.

To verify the superiority of DA-GAN as well as the contribution of each component, we also compare the qualitative results produced by the vanilla GAN (3), Apple GAN (8), BE-GAN (1), and three variations of DA-GAN in terms of w/o $\mathcal{L}_{adv}$, $\mathcal{L}_{ip}$, $\mathcal{L}_{pp}$ in each case, repectively. As shown in Figure. 4, inference without $\mathcal{L}_{ip}$ deviates from the true appearance seriously, and the synthesis without $\mathcal{L}_{adv}$ tends to be very blurry, while the results without the $\mathcal{L}_{pp}$ sometimes show blurry and unnatural effect with strange artifacts / color involved. Compared with vanilla GAN (3), Apple GAN (8) and BE-GAN (1), which all fail with poses larger than $60°$, our DA-GAN presents a good identity preserving quality while producing photorealistic synthesis.

## 6  Verification result analysis for IJB-A Split1

For face verification, after computing the similarities for all pairs of probe and reference sets, we sort the resulting list. Each row represents a probe and reference template pair. The original templates within IJB-A (7) contain from one to dozens of media. Up to eight individual media are shown, with

the last space showing a mosaic of the remaining media in the template. Between the templates are the template IDs for probe and reference as well as the best matched and best non-matched similarities. Figure. 5 shows the best matched cases. In the top-30 scoring correct matches, we immediately note that every reference template contains dozens of media. The probe templates either contain dozens of media or one medium that matches well. Figure. 6 illustrating the best non-matched cases shows the most certain non-mates, again often involving large templates with enough guidance from the relevant information of the same subject. Figure. 7 shows the worst matched cases, representing failed matching. The thirty lowest matched results from single-medium probe sets are all under extremely challenging unconstrained conditions. These extremely difficult cases cannot be solved even using the specific operations designed in our "recognition via generation" framework. Figure. 8 illustrating the worst non-matched cases highlights the understandable errors, representing impostors in challenging modalities.

## 7 Identification result analysis for IJB-A Split1

For face identification, Figure. 9 $1^{st}$-column shows the query images from probe templates. Figure. 9 column 2-6 show the corresponding top-5 queried gallery templates. For each template, we provide template ID, subject ID and similarity score. As can be seen, our approach always performs successful searching in Rank1, which well proved the effectiveness of our DA-GAN based method for generic transfer learning and face-centric analysis. It would be interesting to apply DA-GAN for other transfer learning applications in the future.

## Acknowledgement

The work of Jian Zhao was partially supported by China Scholarship Council (CSC) grant 201503170248.

The work of Jiashi Feng was partially supported by National University of Singapore startup grant R-263-000-C08-133, Ministry of Education of Singapore AcRF Tier One grant R-263-000-C21-112 and NUS IDS grant R-263-000-C67-646.

We would like to thank Junliang Xing (Institute of Automation, Chinese Academy of Sciences), Hengzhu Liu, and Xucan Chen (National University of Defense Technology) for helpful discussions.

## Footnotes

*Homepage: https://zhaoj9014.github.io/.

†Jian Zhao and Zhecan Wang were interns at Panasonic R&D Center Singapore during this work.

[3]ReLU is short for Rectified Linear Units (5).

[4]BN is short for Batch Normalization (6).

[5]Ele-Sum is short for element-wise summation.

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

Figure 2: Refined results of DA-GAN under various poses with yaw angles ranging from $-90°$ to $-10°$ at a stride of $10°$.

Figure 3: Refined results of DA-GAN under various poses with yaw angles ranging from $+10°$ to $+90°$ at a stride of $10°$.

Figure 4: Qualitative result comparison of DA-GAN with state-of-the-art GANs and three different network settings.

Figure 5: Verification results analysis for best matched cases on IJB-A (7) split1.

Figure 6: Verification results analysis for best non-matched cases on IJB-A (7) split1.

Figure 7: Verification results analysis for worst matched cases on IJB-A (7) split1.

Figure 8: Verification results analysis for worst non-matched cases on IJB-A (7) split1.

Figure 9: Identification results analysis on IJB-A (7) split1.