[Reviews · NeurIPS 2017]

Reviewer 1



This work uses GANs to generate synthetic data to use for supervised training of facial recognition systems. More specifically, they use an image-to-image GAN to improve the quality of faces generated by a face simulator. The simulator is able to produce a wider range of face poses for a given face, and the GAN is able to refine the simulators output such that it is more closely aligned with the true distribution of faces (i.e. improve the realism of the generated face) while maintaining the facial identity and pose the simulator outputted. They show that by fine tuning a facial recognition system on this additional synthetic data they are able to improve performance and outperform previous state of the art methods. Pros: - This method is simple, apparently effective and is a nice use of GANs for a practical task. The paper is clearly written Cons: - My main concern with this paper is regarding the way in which the method is presented. The authors term their approach "Dual Agent GANS" and seem to claim a novel GAN architecture. However, it is not clear to me what aspect of their GAN is particularly new. The "dual agent"aspect of their GAN comes from the fact that they have a standard adversarial term (in their case the BE-GAN formulation) plus a cross entropy term to ensure the facial identity is preserved. But previous work (e.g. InfoGAN, Auxiliary classifier GAN) have both also utilized a combination of "heads". So it seems odd to me that the authors are pushing this work as a new GAN architecture/method. I realize it's very trendy these days to come up with a slightly new GAN architecture and give it a new cool name, but this obfuscates the contributions. I think this is an interesting paper from perspective of using GANs in a data augmentation pipeline (and certainly their particular formulation is tailored to the task at hand) but I do not like that the authors appear to be claiming a new GAN method. - Since I think the main contribution of this paper is a data augmentation technique for facial recognition systems, it'd be good to see > 1 dataset explored. Some additional comments/questions: - In eq. 8, do you mean to have a minus sign in the L_G term? - What was the performance of the network you trained before fine tuning? i.e. how much improvement comes from this technique vs. different/better architectures/hyper-parameters/etc. compared to other methods

Reviewer 2



This paper presents a method for augmenting natural face data by 3D synthesis which does not suffer from overfitting on artifacts. The approach uses a GAN network to filter synthesized images so as to automatically remove artifacts. The paper shows that the approach provides a significant boost over a state-of-the-art model on the IJB 'faces in the wild' dataset: reducing the errors by about 25%. The idea of augmenting natural images using 3D models is not new. However, real gains over state-of-the-art performance have not materialized due to the models overfitting on the artifacts of the 3D synthesis process. The authors prove that argument by showing that adding unfiltered augmented data to the baseline model actually degrades performance. I believe this paper shows a promising approach to solve this issue that I have not seen elsewhere so far. The GAN filter uses an original dual-agent discriminator trained with a loss combining the Wasserstein distance with boundary equilibrium regularization and an identity preserving loss. The paper is written clearly, the math is well laid out and the English is fine. I think it makes a clear contribution to the field and should be accepted.

Reviewer 3



This paper proposed a architecture called DA-GAN, where the dual agents focus on discriminating the realism of synthetic profile face images and perceiving the identity information. This work is technically sound and the novelty of this work lies in two points. Firstly, the simulator is designed for realism refinement in DA-GAN inputs. Secondly, the dual agents are combined for embedding data distribution and domain prior of faces. Unfortunately, the overall framework is over-reliant on BEGAN, which leads to much room for further improvement. I have few reasons to believe that most of performance improvement comes from the simulator's face ROI extraction and face landmark detection processing. Correspondingly, the adversarial training part focuses more on integrating identity consistency loss on standard generative authenticity loss by the training method of BEGAN. I am highly recommended the authors conduct experiments to compare the impact of each part whether exist or not. In this way, the advantages of this method can be better identified by the reviewers. The article structure is clear, writing is good and no grammatical errors are found. This method gets good performance improvement in both comparable datasets. However, the whole paper still gives me a faint feeling that the core ideas are not very original and not very solid, so it is safe to say that this paper can but not to a large extent promote more innovative ideas.